# Efficient Elliptic Curve Operators for Jacobian Coordinates

Wesam Eid [1,2,*,†], Turki F. Al-Somani [1] and Marius C. Silaghi [2]

1 Department of Computer Engineering, Umm Al-Qura University, Mecca 21955, Saudi Arabia
2 Department of Computer Engineering and Sciences, Florida Institute of Technology, Melbourne, FL 32901, USA
* Correspondence: wneid@uqu.edu.sa
† Much of the work was performed when Wesam Eid was located at the Florida Institute of Technology.

**Abstract:** The speed up of group operations on elliptic curves is proposed using a new type of projective coordinate representation. These operations are the most common computations in key exchange and encryption for both current and postquantum technology. The boost this improvement brings to computational efficiency impacts not only encryption efforts but also attacks. For maintaining security, the community needs to take note of this development as it may need to operate changes in the key size of various algorithms. Our proposed projective representation can be viewed as a warp on the Jacobian projective coordinates, or as a new operation replacing the addition in a Jacobian projective representation, basically yielding a new group with the same algebra elements and homomorphic to it. Efficient algorithms are introduced for computing the expression $Pk + Q$ where $P$ and $Q$ are points on the curve and $k$ is an integer. They exploit optimized versions for particular $k$ values. Measurements of the numbers of basic computer instructions needed for operations based on the new representation show clear improvements. The experiments are based on benchmarks selected using standard NIST elliptic curves.

**Keywords:** postquantum technology; projective coordinate; Jacobian projective coordinates





## 1. Introduction

Secure Internet-based communications rely on public-key cryptography, which allows entities to communicate without the need for sharing confidential material in advance. Elliptic curve cryptography (ECC), proposed in 1985, is still a predominant type of public-key cryptography [1]. ECC is commonly used for encrypted emails, online banking, secure e-commerce websites, digital signatures, and other data transfer applications where the size of the storage space for public keys is an issue. Breaching these applications would have significant effects on society. The adoption of ECC has been accelerated by recommendations from an array of standardization entities, including NIST, IETF, and ANSI (NIST, 2016). Compared to competitors like RSA and Elgamal, elliptic curve cryptography introduced some of the most efficient public key cryptosystems (PKC) for desirable security. While there are known quantum and classical attacks that breach cryptographic protocols based on supersingular isogeny graphs (SIGs), the supersingular isogeny Diffie–Hellman (SIDH) technique is based on elliptic curves and has no known quantum-based attacks [2].

Quantum computer developments menace to break classic elliptic curve and factoring techniques for public-key cryptography. Supersingular isogeny Diffie–Hellman (SIDH) key exchange is one of the postquantum cryptographic algorithms that can offer secure key exchanges between communicating entities over insecure communication channels [3,4]. The core operations for SIDH are the computation of the isogeny and of its kernel. Basically, Velu's formula is used to compute the isogeny, and the $P + k[Q]$ formula is used to compute the kernel, where $P$ and $Q$ are points on the curve and $k$ is the secret key that is generated by both parties [5]. The complexity of SIDH relies on the difficulty of finding isogenies other than computing the scalar multiplication in the kernel formula. Thus, speeding

up elliptic curve (EC) computations will not only benefit the applications that rely on ECC, but also has an effective impact on the postquantum cryptosystem SIDH. Moreover, attacks are not necessarily limited to independent calculations and might also be based on analyzing hardware behaviors such as with the side-channel attack (SCA). As attackers analyze electrical power consumption patterns, which differ between performing point addition or point doubling, they can recover the secret key. Therefore, the development criteria for EC algorithms and systems can include aspects other than speed. For instance, the addition in the Montgomery coordinate system [6] is resistant to such attacks, being similar to doubling. A drawback of the Montgomery coordinate system is that it is slower than other coordinate systems such as projective and Jacobian [7].

Here, we provide effective algorithms supporting improvements to EC systems in several ways, both in terms of speed-up and resistance to side-channel attacks.

Unlike with other coordinate systems, the original affine operations with the Weierstrass elliptic curve form [7] require computing an inverse each time we perform point doubling or addition, i.e., at every iteration of common fast EC scalar multiplication algorithms. In general, finding inverses is much slower than big integer multiplication. Thus, as commonly done with projective approaches, a main goal of our work is to eliminate inverses. Our first contribution is to compensate the previous abscissa $x$ and ordinate $y$ equations in the higher doubling orders formulas and find a better common factor between all slope's denominators in order to result in a single inverse for each algorithm. In addition, we apply a term regrouping step (referred to as *labeling*) to minimize the number of multiplications. By using these methods, we contribute an efficient way to compute higher doubling orders with an algorithm involving a single inverse. Moreover, we follow the same steps to find intermediate scalar multiplication algorithms, multiplying an EC point P with scalars up to $34P$.

We contribute a new coordinate system using only $x$ and $y$ coordinates and performing a single inverse along the secret key size. The new coordinate system is shown to have competitive properties when compared with other coordinate systems. It can also be seen as a set of new and efficient operators in the Jacobian coordinate space, or on a warped version thereof. Subsequently, we introduce two different competing general scalar EC multiplication algorithms and compare between them using multiple coordinate systems illustrating the strengths of the new proposal.

## 2. Background

The most popular forms of public-key cryptography for current applications have increasingly been based on elliptic curves (ECs) [8,9]. With elliptic curve cryptography (ECC), messages and secrets are mapped to points on an elliptic curve, and specific point-doubling and point-addition operations define transitions between points. Scalar point multiplication uses such a sequence of point-doubling and point-addition operations to optimize repeated addition:

$$Q = [k]P = \underbrace{P + P + \ldots + P}_{k}.$$

Cryptosystems based on ECs rely on the difficulty of solving the elliptic curve discrete log (ECDL) problem. Namely, for elliptic curves with points $P$ of large order and large $k$ numbers, given the points $Q$ and $P$ in the previous equation, it is hard to determine the scalar multiple $k$. However, with the expected emergence of quantum computers [10], in the near future, cryptosystems whose security relies on the difficulty of ECDL will no longer be safe, since the scalar multiple may be easily recovered using Shor's algorithm [11]. Other quantum resilient schemes have been proposed. Furthermore, postquantum cryptosystems such as supersingular isogeny Diffie–Hellman (SIDH) are slow techniques, and speeding up their elliptic curve computation is a significant goal.

The core operation for ECC is the scalar multiplication $[k]P$ whose computation speed is seen as key to improving ciphers. For instant, in [12] Eisentrager et al. proposed a

method for computing the formula $S = (2P + Q)$. Their improved procedure saved a field multiplication, when compared to the original algorithm. Later, Ciet et al. [13] introduced a faster method for computing the same formula when a field inversion costs more than six field multiplications. Furthermore, they introduced an efficient method for computing point tripling. A mixed-powers system of point doubling and tripling for computing the scalar multiplication was represented later by Dimitrov et al. [14]. In [15], Mishra et al. presented an efficient quintuple formula ($5P$) and introduced a mixed-base algorithm with doubling and tripling. A further development was introduced by Longa and Miri [16] by computing an efficient method for tripling and quintupling mixed with a differential addition. They proposed an efficient multibase nonadjacent representation (mbNAF) to reduce the cost. In [16], the same authors presented a further optimization in terms of cost for computing the form $dP + Q$. They succeeded in implementing the previous forms of mixed double-and-add algorithm by using a single inversion when applying a new precomputation scheme. More recently, Purohit and Rawat [17] used a multibase representation to propose an efficient scalar multiplication algorithm of doubling, tripling, and septupling, restricted on a non supersingular elliptic curve defined over the field $F_{2^m}$. In addition, they compared their work with other existing algorithms to achieve a better representation in terms of cost. Therefore, speeding up the scalar multiplication computation in parallel with reducing the cost is a critical task. We present a new methodology to compute elliptic curve operations with more general forms of the type $mP + nQ$, where $m$ and $n$ are small integers, aiming for a faster implementation with the lowest cost among currently known algorithms using only one inversion.

Among all applications based on EC, the highest benefit from our work concerns the postquantum cryptosystem, supersingular isogeny Diffie–Helman (SIDH). Its main weakness is the slow elliptic curve computation speed. For elliptic curve schemes, the computation speed-up also favors attacks, which can however be compensated by increasing the size of the key. Isogeny-based cryptography also utilizes points on an elliptic curve, but its security is instead based on the difficulty of computing isogenies between elliptic curves. An isogeny can be thought of as a unique algebraic mapping between two elliptic curves that satisfy the group law. An algorithm for computing isogenies on ordinary curves in subexponential time was presented by Childs et al. [18], rendering the use of cryptosystems based on isogenies on ordinary curves unsafe in the presence of quantum computers. However, there is no known algorithm for computing isogenies on supersingular curves in subexponential time.

The core operations for SIDH is computing the isogeny using Velu's formula, and its kernel using the $P + k[Q]$ formula, where $P$ and $Q$ are points on the curve and $k$ is the secret key [5]. This operation must be performed in both phases of SIDH. First, this happens in the key generation phase, where the point is known in advance. In this case, one can construct a lookup table that contains all doubles of point $Q$ and reuse any of them when it is needed. Second, in the key exchange phase, where the point $Q$ is variable, we can apply our mixed-base representation (up to 32) in order to speed up the calculations, given that all mixed-base formulas are implemented with a single inversion.

According to Gutub, there are various ways to apply elliptic curves in applications of cryptography [19]. He studied how the algorithm utilized for calculating $nP$ from $P$ was based on the binary representation of $n$, in efficient and practical hardware implementations [19]. That is, the binary algorithm scanned the bits of $n$ and doubled the point $Q$ for a number of $k$-times [19]. Gutub further highlighted how the extra operation of point addition ($Q + P$) was essential, being performed in every case that a particular bit of $n$ was found [19].

### 2.1. Weierstrass Elliptic Curve

This section represents the equations of the original work that we compare our algorithm with. We consider elliptic curves over $\mathbb{Z}_p$, where $p > 3$. Such a curve, in the short Weierstrass form in the affine plan, is the set of all pairs $(x, y) \in \mathbb{Z}_p$ which fulfill:

$$y^2 \equiv x^3 + a \cdot x + b \ (mod \ p) \tag{1}$$

For $P = (x_P, y_P)$ and $Q = (x_Q, y_Q)$, one can compute $P + Q$ by using the following equations, where the computation of $\lambda$ differs based on two disjoint cases [20].

In the case of an addition where $P \neq Q$:

$$\lambda = \left(\frac{y_Q - y_P}{x_Q - x_P}\right) \ mod \ p \tag{2}$$

$$x_R = \lambda^2 - x_P - x_Q \ mod \ p$$

In the case of computing $2 * P$ (doubling of order one) where $P$ has coordinates $(x_1, y_1)$:

$$\lambda = \left(\frac{3x_1^2 + a}{2y_1}\right) \ mod \ p \tag{3}$$

$$x_2 = \lambda^2 - 2x_1 \ mod \ p \tag{4}$$

$$y_2 = \lambda(x_1 - x_2) - y_1 \ mod \ p \tag{5}$$

where $\lambda$ is the slope of the tangent through $P$, and $x_2$ and $y_2$ are the affine coordinates after doubling $P$ one time. While a two-dimensional projective space can also be used for computations in the Weierstrass form, here, we focus on computations in the affine plan.

### 2.2. Projective

Projective coordinates are another way of representing an elliptic curve. The elliptic curve $\Gamma$ can be described by another equation, in the projective space $P^2$. That is, the polynomial defines a curve in the projective space $P^2$, which is also known as a Weierstrass Equation [21]:

$$\Gamma : Y^2 Z + a_1 XYZ + a_3 YZ^2 = X^3 + a_2 X^2 Z + a_4 XZ^2 + a_6 Z^3$$

According to Smart [21], a definition of a projective n-dimensional space $P^n$ over a field F is:

- the set of $(n + 1) - tuples \ (x_0, \ldots, x_n) \in F^{n+1}$
- where at least one $x_i$ per tuple does not equal 0, and;
- where an equivalence relation between two tuples $(x_{0,1}, ..., x_{n,1})$ and $(x_{0,2}, ..., x_{n,2})$ in $P^n$ holds if $\exists U \in F$ such that $\forall i, x_{i,1} = U x_{i,2}$.

We note that a more general definition would replace the third condition with: $\forall i, x_{i,1} = g_i(U) x_{i,2}$ for some bijective function $g_i$.

The equivalence class of $\{U(x_0, \ldots, x_n), \ U \in F\}$ is denoted by $[x_0, \ldots, x_n]$, where these $x_0, \ldots, x_n$ are known as the homogeneous coordinates of that point [21]. Projective coordinates are useful in cases where there is a need to eradicate the performance of costly inversion operations [19].

Higuchi and Takagi [22] and Okeya et al. [23] noted how randomized projective coordinates on a Montgomery-form elliptic curve are effective in securing systems against side-channel attacks. For example, Okeya et al. [23] recommended a scalar multiplication method that does not incur a higher computational cost for randomized projective coordinates of the Montgomery form of elliptic curves.

Homogeneous projective coordinates correspond to the two-dimensional space through the substitution $x = X/Z$ and $y = Y/Z$, so that the general Weierstrass form equates to:

$$E : Y^2Z + a_1XYZ + a_3YZ^2 = X^3 + a_2X^2Z + a_4XZ^2 + a_6Z^3.$$

Jacobian projective coordinates [24–26] are obtained by substituting $x = X/Z^2$ and $y = Y/Z^3$, so that the general Weierstrass form equates to:

$$E : Y^2 + a_1XYZ + a_3YZ^3 = X^3 + a_2X^2Z^2 + a_4XZ^4 + a_6Z^6.$$

With the use of a projective coordinates approach, the attacker is unable to predict the appearance of a specific value when the projective coordinates are randomized [22,23].

Specifically, Higuchi and Takagi [22] proposed a fast addition algorithm on an elliptic curve over GF($2^n$) using projective coordinates:

$$x = X/Z \quad y = Y/Z^2$$

According to Higuchi and Takagi [22], the above projective coordinates have less multiplications than the previously known fastest algorithm [27].

## 3. Affine Recomputation of Multistage Doubling

When computing a scalar multiplication of elliptic curve points $P$ using fast algorithms inspired from Horner's rule, it is common to need operations of the type $kP$, that we refer here as the $k$th-order double of $P$.

In this section, we illustrate how to find a higher-order double independently, without going through all the steps that are required for the original affine coordinates algorithm.

We refer to $2^kP$ as the $k$th double of $P$, and to $kP$ as the $k$th-order double of $P$. We denote by $N_{x_k}$ and $N_{y_k}$ the numerators of the $x$ and $y$ coordinates of the $k$th-order double ($kP$), which are denoted $x_k$ and $y_k$, respectively. Namely, we rewrite $x_k = \frac{N_{x_k}}{U_k^2}$ and $y_k = \frac{N_{y_k}}{U_k^3}$, where $k$ is the order of the desired double, and $U_k$ denotes the corresponding added projective parameter.

When we compute $4P$, first we find $N_{x_2}$ and $N_{y_2}$ as the numerators of $x_2$ and $y_2$, the $x$ and $y$ coordinates of the first double ($2P$), respectively.

For this, substitute the value of $\lambda$ in Equation (3) in both equations of $x$ and $y$ coordinates, then multiply the transformed Equations (4) and (5) with $(2y_1)^2$, for denominators with $U_2 = 2y_1$. The obtained $N_{x_2}$ and $N_{y_2}$ expressions are,

$$N_{x_2} = \left(3x_1^2 + a\right)^2 - 2x_1(2y_1)^2 \ mod \ p \tag{6}$$

$$N_{y_2} = \left(3x_1^2 + a\right)\left(x_1(2y_1)^2 - N_{x_2}\right) - 2y_1^2 \left(2y_1\right)^2 mod \ p \tag{7}$$

We replace the variables $x_2$ and $y_2$ in the second double slope, getting:

$$\lambda_4 = \frac{3x_2^2 + a}{2y_2} \ mod \ p$$

$$\lambda_4 = \frac{3\left(\frac{N_{x_2}}{U_2^2}\right)^2 + a}{2\frac{N_{y_2}}{U_2^3}} \ mod \ p$$

Note that $U_2$ is the denominator of the $(2P)$ slope $\lambda_2 = \lambda$. Now, we eliminate the inverses by amplifying the fraction of $\lambda_4$ with $\frac{U_2^4}{U_2^4}$,

$$\lambda_4 = \frac{3N_{x_2}^2 + aU_2^4}{2N_{y_2}U_2} \ mod \ p \tag{8}$$

For simplicity, we consider,

$$W_4 = 3N_{x_2}^2 + aU_2^4 \ mod \ p \tag{9}$$

$$q_4 = 2N_{y_2} \ mod \ p$$

The new denominator of the obtained slope $\lambda_4$ is:

$$U_4 = q_4 U_2 \ mod \ p \tag{10}$$

Then, we substitute the new slope equation in the $x_4$ and $y_4$ equations,

$$x_4 = \lambda_4^2 - 2x_2 \ mod \ p$$

$$x_4 = \left(\frac{W_4}{U_4}\right)^2 - 2\frac{N_{x_2}}{U_2^2} \ mod \ p$$

Eliminating the inverses in the $x_4$ equation by bringing to a common denominator and amplifying the obtained fraction with the value of $U_4^2$ where we recall from Equation (10) that,

$$U_4 = (2y_1) \ q_4$$

We get,

$$U_4^2 \ x_4 = W_4^2 - 2N_{x_2}q_4^2 \ mod \ p$$

$$x_4 = \frac{W_4^2 - 2N_{x_2}q_4^2}{U_4^2} \ mod \ p \tag{11}$$

where to match

$$x_4 = \frac{N_{x_4}}{U_4^2} \ mod \ p \tag{12}$$

We obtain:

$$N_{x_4} = W_4^2 - 2N_{x_2}q_4^2$$

The same steps are applied in order to find and simplify $y_4$

$$y_4 = \lambda_4(x_2 - x_4) - y_2 \ mod \ p$$

$$y_4 = \frac{W_4}{U_4}\left(\frac{N_{x_2}}{U_2^2} - \frac{N_{x_4}}{U_4^2}\right) - \frac{N_{y_2}}{U_2^3} \ mod \ p$$

Then, we amplify $y_4$ by $U_4^3$

$$U_4^3 \ y_4 = W_4\left(N_{x_2}q_4^2 - N_{x_4}\right) - N_{y_2}q_4^3 \ mod \ p$$

$$y_4 = \frac{W_4\left(N_{x_2}q_4^2 - N_{x_4}\right) - N_{y_2}q_4^3}{U_4^3} \ mod \ p \tag{13}$$

where to match

$$y_4 = \frac{N_{y_4}}{U_4^3} \mod p \tag{14}$$

We obtain

$$N_{y_4} = W_4 \left( N_{x_2} q_4^2 - N_{x_4} \right) - N_{y_2} q_4^3$$

Furthermore, these equations can be generalized for any doubling order. By using this form, one can compute $N_{x_n}$ and $N_{y_n}$ and then replace all the variables in the equation that are related to the order of the desired double in order to perform any advanced double directly (direct doubling). Computing the previous $W_n$'s, $U_n$'s, $N_{x_n}$'s, and $N_{y_n}$'s is required but having the $N_{x_n}$ and $N_{y_n}$ formulas, the computations can be done smoothly.

Here is the general form that performs any doubling:

$$W_n = 3N_{x_{n/2}}^2 + aU_{n/2}^4 \mod p \tag{15}$$

$$q_n = 2N_{y_{n/2}} \mod p$$

$$U_n = q_n U_{n/2} \mod p \tag{16}$$

$$x_n = \frac{W_n^2 - 2N_{x_{n/2}} q_n^2}{U_n^2} \mod p \tag{17}$$

$$x_n = \frac{N_{x_n}}{U_n^2} \mod p \tag{18}$$

$$N_{x_n} = W_n^2 - 2N_{x_{n/2}} q_n^2 \tag{19}$$

$$y_n = \frac{W_n \left( N_{x_{n/2}} q_n^2 - N_{x_n} \right) - N_{y_{n/2}} q_n^3}{U_n^3} \mod p \tag{20}$$

$$y_n = \frac{N_{y_n}}{U_n^3} \mod p \tag{21}$$

$$N_{y_n} = W_n \left( N_{x_{n/2}} q_n^2 - N_{x_n} \right) - N_{y_{n/2}} q_n^3 \tag{22}$$

where $n$ is the order of the double and $n/2$ is assigned to the previous power of 2 double.

*Numerical Examples*

In this section, we use the cyclic group of points on the next elliptic curve $E$, where the order of $E$ is 19 [20]:

$$E : y^2 \equiv x^3 + 2 \cdot x + 2 \mod 17 \tag{23}$$

It is described by the following equations [20]:

$2P = (5,1) + (5,1) = (6,3)$      $11P = (13,10)$
$3P = 2P + P = (10,6)$      $12P = (0,11)$
$4P = (3,1)$      $13P = (16,4)$
$5P = (9,16)$      $14P = (9,1)$
$6P = (16,13)$      $15P = (3,16)$
$7P = (0,6)$      $16P = (10,11)$
$8P = (13,7)$      $17P = (6,14)$
$9P = (7,6)$      $18P = (5,16)$
$10P = (7,11)$      $19P = \mathcal{O}$

As we see, it goes from the primitive element $P = (5,1)$ to $19P$, which represents the identity element, then flips to P again as it is the characteristic of any cyclic group. We use this curve in all our numerical example sections at the end of each algorithm.

Let $P = (5,1)$ to exemplify our direct doubling algorithm. First, we compute $N_{x_1}$ and $N_{y_1}$ that are related to the point $2P = (6,3)$, then we apply another four iterations in order to compute the point $32P \bmod 17$ that is equivalent to the point $13P = (16,4)$.

$$N_{x_2} = (3x_1^2 + a)^2 - 2x_1(2y_1)^2 \ mod \ p$$

$$N_{x_2} = (3(5)^2 + 2)^2 - 2(5)(2(1))^2 \ mod \ 17$$

$$N_{x_2} = 13 - 6 \ mod \ 17$$

$$N_{x_2} = 7$$

$$N_{y_2} = (3x_1^2 + a)(x_1(2y_1)^2 - N_{x_2}) - 2y_1^2 \, (2y_1)^2 \ mod \ p$$

$$N_{y_2} = (3(5)^2 + 2)((5)(2(1))^2 - 7) - 2(1)^2 \, (2(1))^2 \ mod \ 17$$

$$N_{y_2} = 9(3 - 7) - 8 \ mod \ 17$$

$$N_{y_2} = 7$$

where $U_2 = 2y_1 = 2$.

Now, we start the first iteration to find the variables $N_{x_4}$, $N_{y_4}$, $W_4$, $q_4$, and $U_4$ that are related to the point $4P$.

$$W_4 = 3N_{x_2}^2 + aU_2^4 \ mod \ p$$

$$W_4 = 3(7)^2 + 2(2)^4 \ mod \ 17$$

$$W_4 = 9$$

$$q_4 = 2N_{y_2} \ mod \ p$$

$$q_4 = 2(7) \ mod \ 17$$

$$q_4 = 14$$

$$U_4 = q_4 U_2 \ mod \ p$$

$$U_4 = 14(2) \ mod \ 17$$

$$U_4 = 11$$

Then, we substitute these values in the $x_4$ and $y_4$ equations, and we get,

$$x_4 = \frac{W_4^2 - 2N_{x_2}q_4^2}{U_4^2} \ mod \ p$$

$$x_4 = \frac{9^2 - 2(7)(14)^2}{2} \ mod \ 17$$

$$x_4 = \frac{6}{2} \ mod \ 17$$

$$x_4 = 3$$

where the inverse of two is nine and $N_{x_4} = 6$.

$$y_4 = \frac{W_4(N_{x_2}q_4^2 - N_{x_4}) - N_{y_2}q_4^3}{U_4^3} \ mod \ p$$

$$y_4 = \frac{9(7(14)^2 - 6) - 7(14)^3}{11^3} \mod 17$$

$$y_4 = \frac{5}{5} \mod 17$$

$$y_4 = 1$$

where the inverse of five is seven and $N_{y_4} = 5$.

Now, the inputs for the next iteration are ready in order to compute the point $8P$.

$$W_8 = 3N_{x_4}^2 + aU_4^4 \mod p$$

$$W_8 = 3(6)^2 + 2(11)^4 \mod 17$$

$$W_8 = 14$$

$$q_8 = 2N_{y_4} \mod p$$

$$q_8 = 2(5) \mod 17$$

$$q_8 = 10$$

$$U_8 = q_8 U_4 \mod p$$

$$U_8 = 10(11) \mod 17$$

$$U_8 = 8$$

Then, we substitute these values in the $x_8$ and $y_8$ equations, and we get,

$$x_8 = \frac{W_8^2 - 2N_{x_4}q_8^2}{U_8^2} \mod p$$

$$x_8 = \frac{14^2 - 2(6)(10)^2}{8^2} \mod 17$$

$$x_8 = \frac{16}{13} \mod 17$$

$$x_8 = 13$$

where the inverse of 13 is 4 and $N_{x_8} = 16$.

$$y_8 = \frac{W_8(N_{x_4}q_8^2 - N_{x_8}) - N_{y_4}q_8^3}{U_8^3} \mod p$$

$$y_8 = \frac{14(6(10)^2 - 16) - 5(10)^3}{8^3} \mod 17$$

$$y_8 = \frac{14}{2} \mod 17$$

$$y_8 = 7$$

where the inverse of two is nine and $N_{y_8} = 14$.

Now, we substitute with the new values of $N_x$, $N_y$, and U in the next iteration equations in order to compute the point $16P$.

$$W_{16} = 3N_{x_8}^2 + aU_8^4 \mod p$$

$$W_{16} = 3(16)^2 + 2(8)^4 \mod 17$$

$$W_{16} = 1$$

$$q_{16} = 2N_{y_8} \bmod p$$

$$q_{16} = 2(14) \bmod 17$$

$$q_{16} = 11$$

$$U_{16} = q_{16}U_8 \bmod p$$

$$U_{16} = 11(8) \bmod 17$$

$$U_{16} = 3$$

Then, we substitute these values in the $x_{16}$ and $y_{16}$ equations, and we get,

$$x_{16} = \frac{W_{16}^2 - 2N_{x_8}q_{16}^2}{U_{16}^2} \bmod p$$

$$x_{16} = \frac{1^2 - 2(16)(11)^2}{3^2} \bmod 17$$

$$x_{16} = \frac{5}{9} \bmod 17$$

$$x_{16} = 10$$

The inverse of nine is two and $N_{x_{16}} = 5$.

$$y_{16} = \frac{W_{16}(N_{x_8}q_{16}^2 - N_{x_{16}}) - N_{y_8}q_{16}^3}{U_{16}^3} \bmod p$$

$$y_{16} = \frac{1(16(11)^2 - 5) - 14(11)^3}{3^3} \bmod 17$$

$$y_{16} = \frac{8}{10} \bmod 17$$

$$y_{16} = 11$$

The inverse of 10 is 12 and $N_{y_{16}} = 8$.

Now, we substitute with the new values of $N_x$, $N_y$, and U in the last iteration equations in order to compute the desired point $32P$.

$$W_{32} = 3N_{x_{16}}^2 + aU_{16}^4 \bmod p$$

$$W_{32} = 3(5)^2 + 2(3)^4 \bmod 17$$

$$W_{32} = 16$$

$$q_{32} = 2N_{y_{16}} \bmod p$$

$$q_{32} = 2(8) \bmod 17$$

$$q_{32} = 16$$

$$U_{32} = q_{32}U_{16} \bmod p$$

$$U_{32} = 16(3) \bmod 17$$

$$U_{32} = 14$$

Then, we substitute these values in the $x_{32}$ and $y_{32}$ equations, and we get,

$$x_{32} = \frac{W_{32}^2 - 2N_{x_{16}}q_{32}^2}{U_{32}^2} \mod p$$

$$x_{32} = \frac{(16)^2 - 2(5)(16)^2}{(14)^2} \mod 17$$

$$x_{32} = \frac{8}{9} \mod 17$$

$$x_{32} = 16$$

where the inverse of nine is two and $N_{x_{32}} = 8$. Further,

$$y_{32} = \frac{W_{32}(N_{x_{16}}q_{32}^2 - N_{x_{32}}) - N_{y_{16}}q_{32}^3}{U_{32}^3} \mod p$$

$$y_{32} = \frac{16(5(16)^2 - 8) - 8(16)^3}{(14)^3} \mod 17$$

$$y_{32} = \frac{11}{7} \mod 17$$

$$y_{32} = 4$$

where the inverse of seven is five and $N_{y_{32}} = 11$.

## 4. Intermediate Operations

As it is important to calculate the binary multiplicative $2^n$ for points $Q$ to compute a large degree isogeny, we enhance the algorithm by finding the intermediate steps such as $3P, 5P, 7P$, etc.

In [28], Subramanya Rao worked on Montgomery curves and found an efficient technique to find point tripling. Simply, we optimize an application of a single double to some point $P$, then perform a point addition. This technique could be applied to all intermediate steps. We present a set of general forms through which we represent the interstitial points up to $31P$.

### 4.1. Fast $2^n Q + P$

As mentioned earlier in the background section, the complexity of the SIDH cryptosystem relies on computing isogenies between points on the elliptic curve. Thus, we performed a further optimization in term of the kernel equation $P + [k]Q$. As we succeeded to perform an advanced exponent of a point on a curve with a single inverse, it would have required an extra inverse for a differential point addition. Therefore, in this section, we introduce an optimization for mixing our advanced doubling equations with the addition and perform it with a single inverse.

The following equations have some variables such as $N_x$, $N_y$, and U that have to be replaced with the variables related to each double.

We substitute the value of th $x$ and $y$ coordinates of the point $2^n P$ in Equations (18) and (21), respectively, in the addition slope equation in (2).

$$\lambda_n = \frac{\frac{N_{y_n}}{U_n^3} - y_1}{\frac{N_{x_n}}{U_n^2} - x_1} \mod p$$

Multiplying with $U_n^3$ to eliminate the inverses,

$$\lambda_{n+m} = \frac{N_{y_n} - y_1 U_n^3}{N_{x_n} U_n - x_1 U_n^3} \mod p \tag{24}$$

$$W_{n+m} = N_{y_n} - y_1 U_n^3 \mod p \tag{25}$$

$$q_{n+m} = N_{x_n} - x_1 U_n^2 \mod p \tag{26}$$

$$U_{n+m} = U_n q_{n+m} \mod p \tag{27}$$

Substituting $\lambda_{n+m}$ in the equations for $x_{n+m}$ and $y_{n+m}$,

$$x_{n+m} = \left(\frac{W_{n+m}}{U_{n+m}}\right)^2 - x_1 - \frac{N_{x_n}}{U_n^2} \mod p$$

Multiplying with $U_{n+m}^2$,

$$x_{n+m} = \frac{W_{n+m}^2 - x_1 U_{n+m}^2 - N_{x_n} q_{n+m}^2}{U_{n+m}^2} \mod p \tag{28}$$

$$x_{n+m} = \frac{N_{x_{n+m}}}{U_{n+m}^2} \mod p$$

Now, we find $y_{n+m}$,

$$y_{n+m} = \frac{W_{n+m}}{U_{n+m}}\left(x_1 - \frac{N_{x_{n+m}}}{U_{n+m}^2}\right) - y_1 \mod p$$

Multiplying with $U_{n+m}^3$,

$$y_{n+m} = \frac{W_{n+m}(x_1 U_{n+m}^2 - N_{x_{n+m}}) - y_1 U_{n+m}^3}{U_{n+m}^3} \mod p \tag{29}$$

$$y_{n+m} = \frac{N_{y_{n+m}}}{U_{n+m}^3} \mod p$$

Numerical Examples

Let $P = (5,1)$, then we apply our $2^n P + P$ algorithm to compute the new $x$ and $y$ coordinates. In this example, we apply $2^2 P + P$ in order to find the point $5P$. We consider the values previously computed in the numerical example of Section 3 for the point $4P$ where
$N_{x_4} = 6$
$N_{y_4} = 5$
$U_4 = 11$
We substitute these values in Equations (28) and (29) and we get,

$$W_5 = N_{y_4} - y_1 U_4^3 \mod p$$

$$W_5 = 5 - 1(11)^3 \mod 17$$

$$W_5 = 0$$

$$q_5 = N_{x_4} - x_1 U_4^2 \mod p$$

$$q_5 = 6 - 5(11)^2 \mod 17$$

$$q_5 = 13$$

$$U_5 = U_4 q_5 \mod p$$

$$U_5 = 11(13) \mod 17$$

$$U_5 = 7$$

$$x_5 = \frac{W_5^2 - x_1 U_5^2 - N_{x_4} q_5^2}{U_5^2} \quad mod \ p$$

$$x_5 = \frac{(0)^2 - 5(7)^2 - 6(13)^2}{(7)^2} \quad mod \ 17$$

$$x_5 = \frac{16}{15} \quad mod \ 17$$

$$x_5 = 9$$

where the inverse of 15 is 8 and $N_{x_5} = 16$.

$$y_5 = \frac{W_5(x_1 U_5^2 - N_{x_5}) - y_1 U_5^3}{U_5^3} \quad mod \ p$$

$$y_5 = \frac{0(5(7)^2 - 16) - 1(7)^3}{(7)^3} \quad mod \ p$$

$$y_5 = \frac{14}{3} \quad mod \ 17$$

$$y_5 = 16$$

where the inverse of three is six and $N_{y_5} = 14$.

### 4.2. Another General Forms

In Section 4.1, we illustrated the importance of computing the intermediate equations for the overall speed-up of our algorithms. Table 1 lists our general forms that we illustrated to implement all the points up to $31P$.

As it is known, the nonadjacent form (NAF) aims to reduce the number of one bit in the binary representation and thus reduce the number of operations; here, we likewise present in Table 2 the mathematical structure that we relied on for representing all points up to $31P$ with the fastest and most efficient possible form.

**Table 1.** Intermediate Algorithms General Forms.

| Forms | Algorithms |
|:---:|:---:|
| $2^n P + P$ | $W_{n+m} = N_{y_n} - y_1 U_n^3$ <br> $q_{n+m} = N_{x_n} - x_1 U_n^2$ <br> $U_{n+m} = U_n q_{n+m}$ <br> $x_{n+m} = \frac{W_{n+m}^2 - x_1 U_{n+m}^2 - N_{x_n} q_{n+m}^2}{U_{n+m}^2}$ <br> $y_{n+m} = \frac{W_{n+m}(x_1 U_{n+m}^2 - N_{x_{n+m}}) - y_1 U_{n+m}^3}{U_{n+m}^3}$ |
| $2^n P + 2P$ | $W_{n+m} = N_{y_n} - 8N_{y_m}^4$ <br> $q_{n+m} = N_{x_n} - 4N_{x_m} N_{y_m}^2$ <br> $U_{n+m} = U_n q_{n+m}$ <br> $x_{n+m} = \frac{W_{n+m}^2 - 4N_{x_m} N_{y_m}^2 q_{n+m}^2 - N_{x_n} q_{n+m}^2}{U_{n+m}^2}$ <br> $y_{n+m} = \frac{W_{n+m}(4N_{x_m} N_{y_m}^2 q_{n+m}^2 - N_{x_{n+m}}) - 8N_{y_m}^4 q_{n+m}^3}{U_{n+m}^3}$ |

**Table 1.** *Cont.*

| Forms | Algorithms |
|---|---|
| $2(nP)$ | $W_{2n} = 3N_{x_n}^2 + aU_n^4$ <br> $q_{2n} = 2N_{y_n}$ <br> $U_{2n} = U_n q_{2n}$ <br> $x_{2n} = \dfrac{W_{2n}^2 - 2N_{x_n} q_{2n}^2}{U_{2n}^2}$ <br> $y_{2n} = \dfrac{W_{2n}(N_{x_n} q_{2n}^2 - N_{x_{2n}}) - N_{y_n} q_{2n}^3}{U_{2n}^3}$ |
| $2(nP) + P$ | $W_{2n+1} = N_{y_{2n}} - y_1 U_{2n}^3$ <br> $q_{2n+1} = N_{x_{2n}} - x_1 U_{2n}^2$ <br> $U_{2n+1} = U_{2n} q_{2n+1}$ <br> $x_{2n+1} = \dfrac{W_{2n+1}^2 - x_1 U_{2n+1}^2 - N_{x_{2n}} q_{2n+1}^2}{U_{2n+1}^2}$ <br> $y_{2n+1} = \dfrac{W_{2n+1}(x_1 U_{2n+1}^2 - N_{x_{2n+1}}) - y_1 U_{2n+1}^3}{U_{2n+1}^3}$ |

**Table 2.** Structure and Cost for All Intermediate Algorithms.

| Algorithms | Structure | Number of Multiplications |
|---|---|---|
| $3P$ | $2P + P$ | 19 |
| $5P$ | $4P + P$ | 28 |
| $6P$ | $4P + 2P$ | 26 |
| $7P/9P$ | $8P \pm P$ | 38 |
| $10P$ | $2(5P)$ | 38 |
| $11P$ | $2(5P) + P$ | 51 |
| $12P$ | $2(6P)$ | 36 |
| $13P$ | $2(6P) + P$ | 49 |
| $14P/18P$ | $2(7/9P)$ | 48 |
| $15P/17P$ | $16P \pm P$ | 48 |
| $19P$ | $2(9P) + P$ | 61 |
| $20P$ | $2(10P)$ | 46 |
| $21P$ | $2(10P) + P$ | 61 |
| $22P$ | $2(11P)$ | 62 |
| $23P/25P$ | $2(12P) \pm P$ | 59 |
| $24P$ | $2(12P)$ | 46 |
| $26P$ | $2(13P)$ | 59 |
| $27P$ | $2(14P) - P$ | 71 |
| $28P$ | $2(14P)$ | 58 |
| $29P$ | $2(15P) - P$ | 71 |
| $30P$ | $2(15P)$ | 58 |
| $31P$ | $32P - P$ | 58 |

## 5. Extraction of Coordinates (EiSi Coordinates)

The EiSi coordinate system can be seen as a modified version of either the affine or Jacobian spaces with different operators. Each point $P_A = (N_{x_A} : N_{y_A} : U_A)$ is represented

in affine coordinates as $(N_{x_A}/U_A^2, N_{y_A}/U_A^3)$. The EiSi space operators also offer faster arithmetic. Similar to the previous projective techniques, this form of elliptic curve is represented with a single inversion at the last iteration.

Let $P_A$ and $P_B$ be points on an elliptic curve then, in affine space,

$$(X_A : Y_A) + (X_B : Y_B) = (X_C : Y_C)$$

At the first iteration, we consider $U_A = U_B = 1$, then we get,

$$(N_{x_A} : N_{y_A}) + (N_{x_B} : N_{y_B}) = \left( \frac{N_{x_c}}{U_c^2} : \frac{N_{y_c}}{U_c^3} \right)$$

where (in case of doubling),

$$N_{x_C} = \left(3x_A^2 + a\right)^2 - 2x_A(2y_A)^2 \ mod \ p \tag{30}$$

$$N_{y_C} = \left(3x_A^2 + a\right)\left(x_A(2y_A)^2 - N_{x_A}\right) - 2y_A^2 \ (2y_A)^2 \ mod \ p \tag{31}$$

$$U_c = 2y_A \ mod \ p \tag{32}$$

Additionally, in case of adding two points after the first iteration, where the base point will be changed, we illustrate a modified version of the point addition algorithm. Let $P_A$ and $P_B$ be a point on the elliptic curve where $(N_{x_A} : N_{y_A} : U_A)$ and $(N_{x_B} : N_{y_B} : U_B)$ are the projective EiSi points representation, respectively. Then, we get,

$$(N_{x_A} : N_{y_A} : U_A) + (N_{x_B} : N_{y_B} : U_B) = (N_{x_C} : N_{y_C} : U_C)$$

In the case $P_1 \neq \pm P_2$ (addition),

$$W_C = N_{y_B}U_A^3 - N_{y_A}U_B^3 \ mod \ p \tag{33}$$

$$q_C = N_{x_B}U_A^2 - N_{x_A}U_B^2 \ mod \ p \tag{34}$$

$$U_C = U_A U_B q_C \ mod \ p \tag{35}$$

$$N_{x_C} = W_C^2 - N_{x_A}U_B^2 q_C^2 - N_{x_B}U_A^2 q_C^2 \ mod \ p \tag{36}$$

$$N_{y_C} = W_C(N_{x_A}U_B^2 q_C^2 - N_{x_C}) - N_{y_A}U_B^3 q_C^3 \ mod \ p \tag{37}$$

In the case $P_A = P_B$ (higher-order doubling), let $P_1 = P_A$. Then, as proved in Section 3, recursively

$$W_n = 3N_{x_{n/2}}^2 + aU_{n/2}^4 \ mod \ p \tag{38}$$

$$q_n = 2N_{y_{n/2}} \ mod \ p \tag{39}$$

$$U_n = q_n U_{n/2} \ mod \ p \tag{40}$$

$$N_{x_n} = W_n^2 - 2N_{x_{n/2}}q_n^2 \ mod \ p \tag{41}$$

$$N_{y_n} = W_n(N_{x_{n/2}}q_n^2 - N_{x_n}) - N_{y_{n/2}}q_n^3 \ mod \ p \tag{42}$$

We rewrite our algorithms to receive $N_{x_n}$, $N_{y_n}$, and $U_n$ instead of the $x_n$ and $y_n$ values. By applying this method, we manage to dispense with computing the inverse at each iteration.

Since all algorithms start with finding the $N_{x_2}$ and $N_{y_2}$ values that are related to the point $2P$, we note some adjustments to these algorithms in terms of their inputs, then we get:

$$N_{x_2} = (3N_{x_{in}}^2 + aU_{in}^4)^2 - 8N_{x_{in}}N_{y_{in}}^2 \ mod \ p \tag{43}$$

$$N_{y_2} = (3N_{x_{in}}^2 + aU_{in}^4)(4N_{x_{in}}N_{y_{in}}^2 - N_{x_2}) - 8N_{y_{in}}^4 \ mod \ p \tag{44}$$

$$U_2 = 2N_{y_{in}} U_{in} \ mod \ p \tag{45}$$

where $N_{x_{in}}$, $N_{y_{in}}$, and $U_{in}$ are the inputs that represent the point $(X_1 : Y_1)$ at the first iteration where $U_{in}$ equals one and $N_{x_2}$, $N_{x_2}$, and $U_2$ are the outputs that represent the point $2P$.

*Numerical Example*

In this section, we use the same cyclic group introduced in Section 3 and consider some of the values previously computed in the previous numerical examples sections in order to illustrate how our new coordinate system finds point doubling and addition correctly with a single inverse along the key size.

Assume we have a key size of four bits that represents the number $10_{10} = (1010)_2$. Then, we apply the left-to-right algorithm in order to compute the new $x$ and $y$ coordinates for the point $10P = (7,11)$.

First, as we scan from left to right we process the second one-bit. Each one-bit is represented by a doubling and addition while each zero-bit is only represented by a doubling. Thus, we double in order to find the point $2P$.

As in Section 3, we consider the values,
$U_2 = 2$
$N_{x_2} = 7$
$N_{y_2} = 7$

Then, we scan the next bit from the left which appears to be 1. Another double-and-add operation is applied and we get $2(2P) + P = 5P$.

KEY:     1     0     1 ...
Operations:     2P     5P ...

As in Section 4.1, we consider the values for $5P$ as well,
$U_5 = 7$
$N_{x_5} = 16$
$N_{y_5} = 14$

Note: $N_{x_2}$, $N_{y_2}$, $N_{x_5}$, and $N_{y_5}$ are computed with no inversion operation.

Now, we scan the last bit which appears to be 0. The doubling operation is applied and we get $2(5P) = 10P = (7,11)$.

KEY:     1     0     1     0
Operations:     2P     5P     10P

$$W_{10} = 3N_{x_5}^2 + aU_5^4 \ mod \ p$$

$$W_{10} = 3(16)^2 + 2(7)^4 \ mod \ 17$$

$$W_{10} = 11$$

$$q_{10} = 2N_{y_5} \ mod \ p$$

$$q_{10} = 2(14) \ mod \ 17$$

$$q_{10} = 11$$

$$U_{10} = q_{10}U_5 \ mod \ p$$

$$U_{10} = 11(7) \ mod \ 17$$

$$U_{14} = 9$$

$$N_{x_{10}} = W_{10}^2 - 2N_{x_5}q_{10}^2 \ mod \ p$$

$$N_{x_{10}} = (11)^2 - 2(16)(11)^2 \ mod \ 17$$

$$N_{x_{10}} = 6$$

$$N_{y_{10}} = W_{10}(N_{x_5}q_{10}^2 - N_{x_{10}}) - N_{y_5}q_{10}^3 \ \ mod \ \ p$$

$$N_{y_{10}} = 11((16)(11)^2 - 6) - (14)(11)^3 \ \ mod \ \ 17$$

$$N_{y_{10}} = 12$$

At the end of the last iteration, the inverse function is applied in order to find the affine coordinates for the point $10P$.

$x_{10} = \frac{N_{x_{10}}}{U_{10}^2} = \frac{6}{13} = 7$

$y_{10} = \frac{N_{y_{10}}}{U_{10}^3} = \frac{12}{15} = 11$ where $13^{-1} = 4$ and $15^{-1} = 8$.

## 6. Fast Multiplication with Mixed-Base Multiplicands

Here follows the description of a few algorithms that can integrate the fast repeated-doubling techniques mentioned so far by applying mixed base multiplicands. With the algorithm $mP + nQ$, one can compute multiplications with scalars up to 31. One can divide $m$'s binary representation into blocks of five bits. In case an obtained block represents one of the unimplemented scalar multiplications, such blocks may be reduced in length.

### 6.1. Double-and-Add Extensions

In Sections 3 and 4, it was shown how to compute all intermediate exponents and mix the doubling with a differential addition with a single inverse. The left-to-right algorithm starts scanning from the left the next one-bit considering that the most significant bit is one. Then, it decides whether it applies doubling or doubling and addition depending on the data being read. For instance, if the first two one-bits represent the binary equivalent $(101)_2$ which is $5_{10}$, the algorithm multiplies the base by four because it was shifted to the left by two bits. Since the last bit scanned is a one, it also applies a differential addition to the point being doubled with the base point. Thus, the implementation is $4Q + Q$. Figure 1 shows a practical example for calculating $Q^{47}$. This technique computes $Q^{47}$ with only four inverses, instead of the eight inverses when performing the original equations. However, by applying our EiSi coordinate system, one can compute the whole key with a single inverse as for the projective technique.

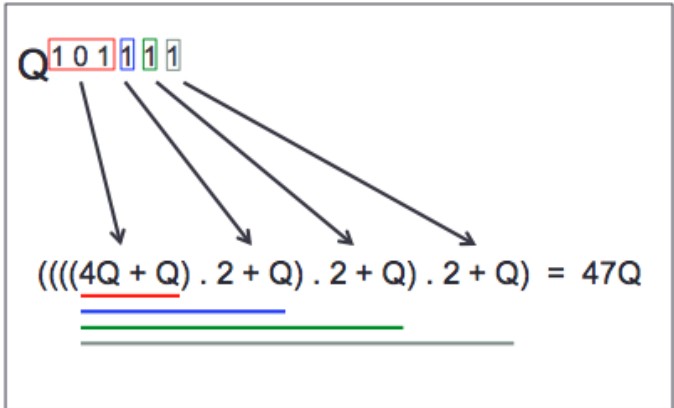

**Figure 1.** Left-to-Right Proposed Algorithm.

In Line 1 of the pseudocode of the double-and-add extensions in Algorithm 1, we apply the DoubleAndAddKnapsack function taking as parameter the counter l that specifies the current bit location, and the base point P to be added at the end. Otherwise, we apply the DoubleKnapsack function that computes the shifting to the left by multiplying the D value with $2^l$.

---

**Algorithm 1:** Double-and-Add Extensions

---

**procedure** *MultiplyL2RKnapsack(k, P)* **do**

  **if** $|k| \leq 0$ *or* $k_{|k|-1} == 0$ **then** return O;

  ;

  D := P;

  **for** *(int i := |k| − 2; i ≥ 0; )* **do**

    l := 0;

    **do**

      i − −;

      l ++;

    **while** *($k_i$ == 0 and i ≥ 0)*;

    **if** $k_i$ == 1 **then**

1      D := DoubleAndAddKnapsack(D, l, P);

    **else**

2      D := DoubleKnapsack(D, l);

    **end**

  **end**

  **return** D;

**end**

---

*6.2. Fast Multiplication with Base-32 Multiplicands*

Here, we mention a simple special case of an algorithm based on base-32 representations of the multiplicands. Then, for a multiplier of the form $\overline{qr_{32}}$ the computation is implemented as,

$$32(qP) + rP \ mod \ p$$

For the scalar $10{,}150 = 27A6_{16}$, the obtained algorithm is equivalent to:

$$(32(9P) + 29P)(32) + 6P \ mod \ p$$

As noted in the above equation, and similar to the Montgomery curve, the key is indistinguishable and cannot be recognized by a side-channel attack since the algorithm applies point doubling and addition at each iteration regardless of the key bit value. In addition, by applying direct-doubling algorithms, we benefit from the reduced number of point additions, which in fact costs more than point doubling. Moreover, as noted in Section 5, the EiSi coordinate system operates in two modes. The first is when it receives affine $x$ and $y$ coordinates, while the other deals with $N_x$, $N_y$, and U as inputs. The base-32 multiplicands algorithm increases the use of the first mode, which costs fewer multiplications. Thus, we consider a base-32 multiplicands algorithm as one of the most efficient algorithms.

## 7. Results and Experiments

Simulation experiments were performed with a Java implementation of the proposed algorithms. We applied the algorithms on large parameters defined in the standard curves P-521, P-384, P-256, and P-224 from the National Institute of Standards and Technology (NIST). In addition, we picked 10 different keys that were randomly generated with an appropriate size for the $x$ and $y$ coordinates of each curve. Each algorithm was executed multiple times and then we computed the average time taken to increase the accuracy of the calculations. Experimentally, our software implementation was tested on a BeagleBone Black (BBB) System kit [29]. The BBB was equipped with a minimum set of features to allow the user to experience the power of the processor [29]. This system is equipped with one of the ARM Cortex-A8 family, AM3358/9 processor [29].

*7.1. Functions Description and Properties*

In this section, we list the important functions that were used in our software implementation and their properties. Since the EiSi curve receives and returns two different forms of inputs and outputs, (x,y) or ($N_x$:$N_y$:U), we clarify among these characteristics these details.

### 7.1.1. doubling2$^n$N

This special function was designed to receive and return an EiSi point. Basically, it receives the number of doublings of a point and then builds the equation for implementing this doubling. For example, if one wants to compute the point 6$P$, it requires finding the point 2$P$ then 4$P$ in order to fulfill the constructional equation for 6$P$, which is 2$P$ + 4$P$.

### 7.1.2. adv_addN2N_N and adv_subN2N_N

These functions receive and return EiSi points. Briefly, they perform point addition and subtraction between two EiSi points.

### 7.1.3. remi_point

This function receives an affine point and return an EiSi point. In addition, it receives the number of doublings of a point and then builds the equation for implementing this doubling. Mainly, it is used in the base-32 multiplicands algorithm specifically for computing the remainders, where all of them are based on the same base point. It differs from the doubling2$^n$N function, where all the doubling algorithms operators and labels are dependent. Basically, the point 4$P$ cannot be computed without finding the point 2$P$. Moreover, the point 8$P$ cannot be computed without finding the point 2$P$ then 4$P$. For example, if one wants to compute the third double for the base point 8$P$ = (13,7), it is represented in EiSi coordinates as (16:14:8). The remi_point will compute the $N_x$, $N_y$, and U values for the point 2(8$P$) then 4(8$P$) then return the EiSi point of 8(8$P$) = 7$P$ mod 19 that is represented as (0:14:2).

### 7.1.4. remi_func

This function works as a control for the remi_point function. The remi_func has architectures of all the doubling algorithms and how they are implemented. Essentially, it has flags to be checked to avoid repeating any previously computed operations. Figures 2 and 3 show practical examples printed from our implementation debug page.

```
2P
Found 2P
Output = (6,3)
-----------------------------
9P = 8P + P
Found 4P
Found 8P
Found 9P
Output = (7,6)
-----------------------------
24P = 16P + 8P
Found 16P
```

**Figure 2.** Computing the point 24P by using remi_func and remi_point functions.

```
29P = 32P − 3P
Found 2P
Found 4P
Found 8P
Found 16P
Found 32P
Found 3P
Found 29P
Output = (7, 11)
─────────────────

11P = 8P + 3P
Found 11P
Output = (13, 10)
```

**Figure 3.** Computing the point $29P$ by using remi_func and remi_point functions.

As we notice in Figure 2, at the second block the 2P parameters were not recomputed and similarly at the third block for 8P. Furthermore, in Figure 3, at the last block, we notice that 8P and 3P were previously computed and we only had to perform a point addition.

### 7.2. Our Work vs. Original

In this section, we compare our algorithms in terms of number of multiplications for our work, base-32 multiplicands and double-and-add, with the original affine algorithm. We implemented the original affine equations with two different algorithms, right-to-left and left-to-right. Table 3 shows the huge differences in the number of multiplications and inversions between these algorithms and our work. On colored background one can see the best results for each benchmark, and they correspond to the Base 32 version of our technique.

**Table 3.** Our Work vs. Original Algorithms Measurements.

| NIST Curve | Algorithm | Number of Operations | |
| --- | --- | --- | --- |
| | | Mult. | Inv. |
| P-521 | RL (original) | 658,514 | 1059 |
| | LR (original) | 478,056 | 778 |
| | DA | 9162 | 1 |
| | Base 32 | 7921 | 1 |
| P-384 | RL (original) | 355,788 | 775 |
| | LR (original) | 259,177 | 569 |
| | DA | 6714 | 1 |
| | Base 32 | 5890 | 1 |
| P-256 | RL (original) | 162,131 | 519 |
| | LR (original) | 115,340 | 378 |
| | DA | 4466 | 1 |
| | Base 32 | 4007 | 1 |
| P-224 | RL (original) | 123,461 | 450 |
| | LR (original) | 88,921 | 332 |
| | DA | 3921 | 1 |
| | Base 32 | 3529 | 1 |

As we notice in Table 3, there is a great difference in the number of multiplications and inversions between our work and the original algorithm as our work is faster by approximately 35 up to 83 times for the key sizes of 224 bits and 521 bits, respectively, when comparing with RL and 25 up to 60 times in the case of LR. Obviously, all these differences are caused by the number of inverse operations that the original algorithm requires for each point doubling or addition operation. Figure 4 translates this difference in a chart where our work appears as a straight line along the *x*-axis.

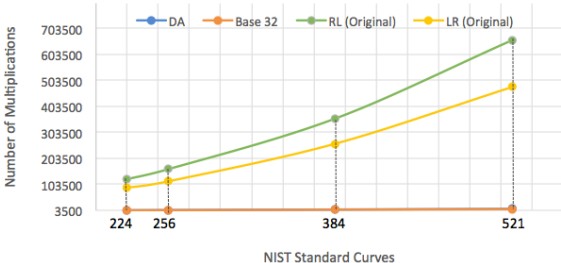

**Figure 4.** Our work vs. Original algorithms in Terms of Number of Multiplications: blue and red lines are superposed.

### 7.3. Eisi Coordinates vs. Others

Here, we compare our work with the other coordinate systems, projective and Jacobian. Table 4 shows a comparison of these algorithms in terms of the number of additions, subtractions, multiplications, divisions, modulo operations, maximum levels of parallelization, and elapsed time for implementing them on the NIST standard curves P-521, P-384, P-256, and P-224.

**Table 4.** EiSi Coordinates vs. Other Coordinates Measurements.

| NIST Curve | Algorithm | Number of Operations | | | | | Time ms |
| --- | --- | --- | --- | --- | --- | --- | --- |
| | | Mult. | Div. | ALUs | Mod. | MaxLs | |
| P-521 | Projective | 14,900 | 315 | 4211 | 17,899 | 9390 | 1035 |
| | Jacobian | 13,312 | 301 | 4197 | 15,821 | 8862 | 884 |
| | Base 32 | 7921 | 296 | 6518 | 12,290 | 7123 | 778 |
| | DA | 9162 | 301 | 7848 | 18,528 | 9924 | 951 |
| P-384 | Projective | 10,901 | 226 | 3078 | 13,107 | 6871 | 554 |
| | Jacobian | 9752 | 222 | 3075 | 11,587 | 6491 | 480 |
| | Base 32 | 5890 | 227 | 4833 | 9113 | 5224 | 421 |
| | DA | 6714 | 222 | 5746 | 13,556 | 7267 | 508 |
| P-256 | Projective | 7236 | 145 | 2043 | 8714 | 4564 | 286 |
| | Jacobian | 6488 | 147 | 2046 | 7708 | 4316 | 261 |
| | Base 32 | 4007 | 150 | 3277 | 6210 | 3463 | 227 |
| | DA | 4466 | 147 | 3820 | 9028 | 4833 | 266 |
| P-224 | Projective | 6356 | 127 | 1796 | 7656 | 4010 | 234 |
| | Jacobian | 5697 | 127 | 1796 | 6773 | 3790 | 215 |
| | Base 32 | 3529 | 132 | 2886 | 5472 | 3041 | 193 |
| | DA | 3921 | 127 | 3354 | 7931 | 4243 | 218 |

As we notice in Table 4, our work which is represented in the last two algorithms is more efficient when it comes to number of multiplications (see entries with colored background). Clearly, the base-32 multiplicands algorithm is the optimal algorithm in this case. Moreover, when we compare by the maximum levels of parallelization, we find that our work outperforms the other coordinates algorithms as well through the base-32 multiplicands algorithm which makes it the optimal algorithm in terms of both factors. Nevertheless, the double-and-add algorithm which represents the original EiSi coordinates appears to be the least efficient in terms of maximum levels; however, together with our direct-doubling technique we outperform all other algorithms in all aspects. Figures 5 and 6 contain graphs that show the comparisons between our work and the other coordinates algorithms in terms of number of multiplications and maximum levels of parallelization with different key sizes.

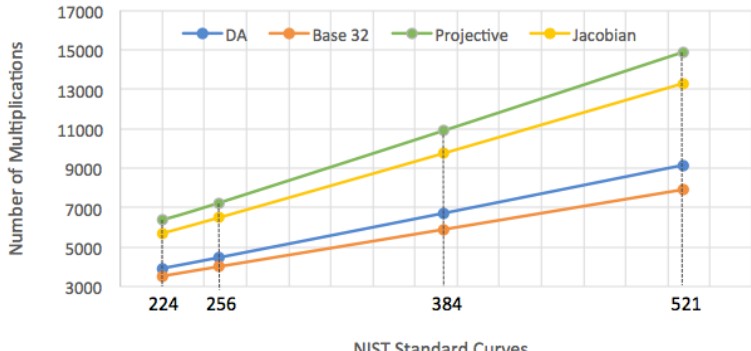

**Figure 5.** Our Work vs. Other Coordinates Algorithms in Terms of Number of Multiplications.

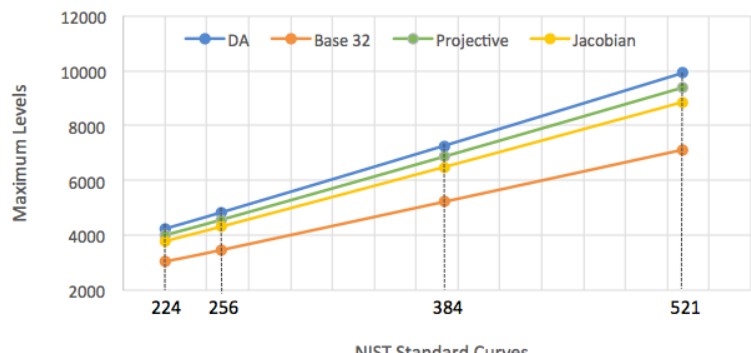

**Figure 6.** Our Work vs. Other Coordinates Algorithms in Terms of Number of Maximum Number of Levels.

As can be seen in Figures 5 and 6, all algorithms are graphed as straight lines of varying slopes, which gives us the opportunity to apply a straight-line equation to any algorithm to predict the expected values, whether it is the number of multiplications or the maximum levels of parallelization at the level of a larger key size. Table 5 lists all the equations related to each algorithm.

**Table 5.** List of Algorithms' Linear Equations.

| Algorithm | Equation: Number of Mult. | Equation: Number of MaxLs |
|---|---|---|
| Base 32 | $y = 14.777x + 220.37$ | $y = 13.762x - 52.445$ |
| DA | $y = 17.658x - 48.287$ | $y = 19.139x - 60.183$ |
| Jacobian | $y = 25.656x - 70.992$ | $y = 17.089x - 52.485$ |
| Projective | $y = 28.795x - 121.85$ | $y = 18.131x - 69.095$ |

Predictably, we apply the equations in Table 5 on two key sizes of the prime numbers of 751 and 1013. Table 6 lists the expected number of multiplications and maximum number of levels.

**Table 6.** Expected number of Multiplications and Maximum Levels with Key of Sizes 751 and 1013.

| Algorithm | Expected Number of Mult. | | Expected Number of MaxLs | |
|---|---|---|---|---|
| | **751** | **1013** | **751** | **1013** |
| Base 32 | 11,317 | 15,189 | 10,282 | 13,888 |
| DA | 13,212 | 17,839 | 14,313 | 19,327 |
| Jacobian | 19,196 | 25,918 | 12,781 | 17,258 |
| Projective | 21,503 | 29,047 | 13,547 | 18,297 |

As it can be seen in Table 6, our work represented in the base-32 multiplicands algorithm maintains its place as the optimal algorithm in terms of number of multiplications and maximum levels of parallelization. Likewise, we find that the Jacobian algorithm outperforms the projective algorithm in terms of the same factors, which led us to another comparison between base-32 multiplicands algorithm and the Jacobian coordinates algorithm to monitor if the difference in performance shrank with the size of the key or continued to increase. Despite the slope values in the straight-line equations that show the differences, we computed the delta value, $\Delta$, which was the difference between the $y$-axis values along the key size. Table 7 shows the comparison between these two algorithms in terms of the same two factors, where

$$\Delta_i = y_n - y_m \tag{46}$$

where $i$ represents the key size and $n$ and $m$ represent the algorithms labels.

**Table 7.** $\Delta$ Values for Base 32 vs. Jacobian.

| Key Size | Number of Mult. | Number of MaxLs |
|---|---|---|
| | **Base 32 vs. Jacobian** | **Base 32 vs. Jacobian** |
| 224 | 2168 | 749 |
| 256 | 2481 | 853 |
| 384 | 3862 | 1267 |
| 521 | 5391 | 1739 |
| 751 | 7879 | 2499 |
| 1013 | 10,729 | 3370 |

We conclude from the $\Delta$ values from Table 7 that our results in both cases showed that the improvement scaled with the size of the input.

*7.4. Number of Multipliers Comparison*

After the tests and comparisons have proven the efficiency of our algorithms and their superiority against other coordinate systems algorithms, in this section we specify the number of multiplication units each algorithm requires to achieve their maximum levels of parallelism. Table 8 shows the number of multipliers per algorithm in the case of a key size of 521.

As it can be seen in Table 8, the appropriate number of multipliers to achieve the highest level of parallelism varies among algorithms. In addition, we note that if we reduce

the number of multipliers a little, we may get a very close result in terms of maximum levels of parallelization. Thus, it led us to another close comparison in which we monitored the behavior of each algorithm in comparison with the others in multiple cases where the number of multipliers was uniform. Table 9 shows another comparison between our optimal algorithm specified in the previous sections compared to the Jacobian algorithm, in terms of the maximum levels (MaxLs) at specific numbers of multipliers.

**Table 8.** The Number of Multipliers Appropriate to Achieve the Highest Level of Parallelism.

| Algorithm | Number of MaxLs | Multipliers |
|-----------|-----------------|-------------|
| DA | 9920 | 3 |
| Base 32 | 7133 | 3 |
| Projective | 9370 | 6 |
| Jacobian | 8862 | 4 |

**Table 9.** Maximum Levels at Different Numbers of Multipliers.

| Multipliers | Algorithm | Number of MaxLs |
|-------------|-----------|-----------------|
| 1 | Base 32 | 7982 |
|   | Jacobian | 13,011 |
| 2 | Base 32 | 7134 |
|   | Jacobian | 8864 |
| 3 | Base 32 | 7133 |
|   | Jacobian | 8863 |
| 4 | Base 32 | 7133 |
|   | Jacobian | 8862 |

As it can be seen in Table 9, our algorithms outperform the Jacobian algorithm in all levels, starting from a single multiplier, where the base-32 algorithm appears to be 63% more efficient, up to four multipliers, where Jacobian reaches its peak while being 24% slower than the base 32 algorithm. In the two-multiplier case, the performance of both algorithms improves significantly as the difference in efficiency becomes almost 24% with the advantage remaining for our work. We also note that the base-32 and Jacobian algorithms become highly ineffective as they continue to increase by one parallel level by increasing the number of multiplication units each time until they reach their peak. At the end, and in all cases, whether we use fewer or more multipliers, the efficiency of our algorithm clearly outweighs the work of other coordinate systems algorithms. Figure 7 shows the graphical representation of the relationship of the number of multipliers with the maximum levels of parallelism.

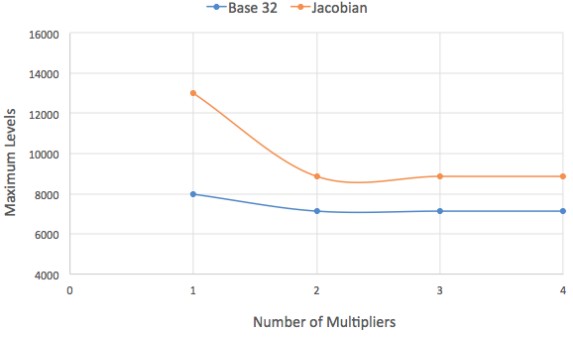

**Figure 7.** Maximum Levels for Different Number of Multipliers.

## 8. Discussion

In 2022, NIST released its report on algorithms resistant to quantum computer attacks, with supersingular isogeny key encapsulation, SIKE, as one of the candidates [2]. The organization clarified in its latest report the advantages of the algorithm in terms of security and key shortness, and in return, its biggest flaw was its performance. Since the SIDH algorithm is based on elliptic curve calculations, our proposal in this paper becomes of great importance and support for this algorithm in the field of quantum computer attacks. Later that year, Castryck and Decru presented an efficient key-recovery attack on the SIDH, based on a "glue-and-split" theorem exploiting the knowledge of the starting curve [30]. A short time later, Damien Robert published a paper claiming to break SIDH in a polynomial time even with a random starting curve [31]. Nevertheless, Fouotsa proposed a countermeasure to the Castryck–Decru attack by applying a mask to the torsion-point images where the attack cannot be valid [32]. Simply, they applied a scalar multiplication $a$ where $a \in Z/BZ^x$ to the points $p$ and $q$. Thus, there will be an impact on performance to maintain security, which makes our contribution more relevant for the application.

## 9. Conclusions

This research proposed optimization methods for computing scalar multiplication in elliptic curves over a prime field, in the short Weierstrass form in the affine plane. The report started by describing a methodology for direct repeated point doubling with a high order, as well as point addition of the form $nP + mQ$ by using a single inversion. These new algorithms were shown to be significantly faster than the original equations. In addition, we developed optimized equations for repeated doubling of higher order than available with comparable current existing algorithms (up to 31).

The second part introduced a new coordinate system, EiSi, with fast algorithms shown to be offering the lowest cost when using only a single inverse. In fact, EiSi shared the same Jacobian space but with different operators. Parallelization opportunities were also highlighted. The evaluation of our implementation indicated that the proposed equations outperformed the other coordinate systems and also provided a significant speed-up when realized in hardware. Moreover, EiSi with the direct repeated doubling technique was proven more efficient in all aspects evaluated, namely the number of multiplications, maximum levels of parallelization, and estimated time.

The base-32 multiplicands algorithm was one of the multiplicands family of algorithms proposed. This algorithm was characterized by its speed and by the low number of parallel levels required to implement it, namely just three multipliers. In addition, the operators of this algorithm kept the key bit values indistinguishable, as they continued the point doubling and addition process according to the Montgomery procedure, making them similarly resistant to side-channel attacks. The base-32 multiplicands algorithm outperformed other coordinates algorithms in all evaluated aspects.

**Author Contributions:** Data curation, W.E., T.F.A.-S and M.C.S.; Formal analysis, W.E., T.F.A.-S and M.C.S.; Funding acquisition, W.E., T.F.A.-S and M.C.S.; Methodology, W.E., T.F.A.-S and M.C.S.; Project administration, W.E., T.F.A.-S and M.C.S.; Software, W.E., T.F.A.-S and M.C.S.; Writing—original draft, W.E., T.F.A.-S and M.C.S.; Writing—review & editing, W.E., T.F.A.-S and M.C.S. All authors have read and agreed to the published version of the manuscript.

**Funding:** Deanship of Scientific Research at Umm Al-Qura University grant 20UQU0026DSR.

**Data Availability Statement:** No data is involved.

**Acknowledgments:** The authors would like to thank the Deanship of Scientific Research at Umm Al-Qura University for supporting this work by grant code: (20UQU0026DSR).

**Conflicts of Interest:** The authors confirm that there is no conflict of interest to declare for this publication.

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
