# Peer review of "Efficient Elliptic Curve Operators for Jacobian Coordinates"

_electronics, doi:10.3390/electronics11193123_

Round 1

Reviewer 1 Report

In this paper, the authors discussed a new type of projective coordinate representation. The proposed projective representation can be viewed as a warp on the Jacobian projective coordinates, or as a new operation replacing the Jacobian projective representation addition and generating a new group with the same algebra elements and homomorphic to it. It is shown that efficient algorithms are introduced for computing the expression pk + Q here and Q are points on the curve and k is an integer, based on optimized versions for particular k values. 

This paper contains new and interesting results to be published after taking the following comments into account

1) The abstract should be rewritten in a more clear way

2) In the introduction section lin2 28 to 35, the authors write ""ECC is commonly used for encrypted emails, online banking, secure ecommerce websites, digital signatures, and other data transfer applications where the size of the storage space for public keys is an issue [........1-3]. Breaching these applications would have significant effects on society [....... 4]. The adoption of ECC has been accelerated by recommendations from an array of standardization entities, including, NIST, IETF, and ANSI (NIST, 2016). Compared to competitors like RSA and Elgamal, elliptic curve cryptography introduced some of the most efficient public key cryptosystems (PKC) for desirable security [.... 5-8].""

Meshal Mohammed Al Anazi, Osama R. Shahin, A Machine Learning Model for the Identification of the Holy Quran Reciter Utilizing K-Nearest Neighbor and Artificial Neural Networks, Information Sciences Letters, Vol. 11, No. 04 (2022) PP: 1093-1102 doi:10.18576/isl/110410

Marghny H. Mohammed, Botheina H. Ali, Ahmed I. Taloba, Self-adaptive DNA-based Steganography Using Neural Networks, Information Sciences Letters, Vol. 8, No. 1  PP: 15-23 (2019) doi:10.18576/isl/080102

K. Thilagavathi, A. Vasuki, A Novel Hyperspectral Image Classification, Technique Using Deep Multi-Dimensional Recurrent Neural Network, Applied Mathematics & Information Sciences, Volume 13, No. 6 PP: 955-963 (2019) doi:10.18576/amis/130608

K. Poongodi, A. K. Sheik Manzoor, Sequential Pattern Mining using RadixTreeMiner Algorithm and Neural Network-Based Classification, Applied Mathematics & Information Sciences, Volume 13, No. S1 PP: 1-16 (2019) doi:10.18576/amis/13S101

P. Shanmugapriya, V. Mohan, T. Jayasankar, Y. Venkataramani, Deep Neural Network based Speaker Verification System using Features from Glottal Activity Regions, Applied Mathematics & Information Sciences, Volume 12, No. 6 PP: 1147-1155 (2018) doi:10.18576/amis/120609

Alaa Zuhir Al Rawashdeh, Asma Rebhi Al Arab, Noura Nasser Alqahtani and Zineb Hadmer, The Sociological Understanding, for Corona Crises and its Reflections on Society: An Inductive Analytical Vision, J. Stat. Appl. Prob. Vol. 10, No. 1 (Mar. 2021), PP:267-286 doi:10.18576/jsap/100123

Abdel-Aty, A.-H., Kadry, H., Zidan, M., Zanaty, E.A., Abdel-Aty, M. A quantum classification algorithm for classification incomplete patterns based on entanglement measure, Journal of Intelligent and Fuzzy Systems, 2020, 38(3), pp. 2817–2822 DOI: 10.3233/JIFS-179566

3) Line 38, the citation is not correct """based attacks [?]. ""'

4) Line 59 to line 79 is very long paragraph and should be divided into two paragraphs 

5) Equation in line 233 should be corrected 

6) The quality of Figure 5: needs to be improved 

7) More discussions are needed on the appropriate number of multipliers to achieve the highest level of parallelism varies between algorithms.

8) In the references list the same format should be used in all references i.e. the year in some references at the end and in others are not 

The misprint in Ref. 14 should be corrected "14. P. K. Mishra and V. Dimitrov, “Efficient quintuple formulas for ellip- tic curves a "

Reviewer 2 Report

Strength:

+ The topic of designing efficient elliptic curve operators for Jacobian Coordinates is important.

+ The methodology is clearly motivated, explained, and the results are promising.

+ The results of the performance evaluation are quite promising.

+ The experiments setting in the standard 641

curves P-521, P-384, P-256 and P-224 are very interesting.

+ The paper is generally well written and organized.

Weakness:

- The (rather advanced) encryption is not well motivated.

- The communication channel security could be compromised by insider attackers if this method is applied in networks.

- Not clear why the Base 32 algorithm is appeared to be 63% 782 more efficient, up to 4 multipliers, where Jacobian reaches its peak with 24% slower than 783 Base 32.

- The overhead experiments are missing.

- The results of the proposed methods are not compared to those of existing works.

- Important security analysis is missing.

Author Response

We have fixed the format issues and rewrote the abstract. We have fixed the citations format as well. Figures were uploaded in better resolusion.  We have added a discussion section to clear why our contribution is important to the post quantum cryptosystem SIDH.

Reviewer 3 Report

In this article, the authors propose a new method to improve the elliptic curve operations for Jacobian Coordinates by reducing the complexity of inversion and multiplication. The authors also provide a new coordinate system based on the Jacobian coordinate. The technical details are clearly explained. But, to improve the quality of the proposals, I suggest the following point:

Minor:

  • In Introduction, the reference in line 38 is missing.

  • In section 2, please use 2.1 and 2.2 instead of a) and b)

  • In section 4, the sub-section 4.1 is missing.

  • In the manuscript, equation number begins with (8) on page 5?!! Please number all the equations.

  • I don’t know why there are so much white spaces in some tables. For example, Table 1 on page 7, Table 2 on page 11, and Table 3 on page 18-19. Please re-do those tables and remove white spaces to make them look better.

  • Why there is two Table 2 (on page 11 and on page 18) and two Table 3 (on page 12 and on page 18-19)? Please number Table properly.

  • Table 3 on page 18-19 spaned over two pages. Please keep the table on only 1 page.

  • Figure 5, 6, 7, and 8 are too small and hard to see. Please enlarge them accordingly.

Major:

  • There is no Figure 1 in the manuscript. Is Figure 1 missing or all the figures are just numbered incorrectly?

  • The authors said that their proposal concerns the SIDH. Therefore, the effect of the provided method on SIDH should be further analyzed.

  • The authors should refer to the newer existing references.

  • The provided algorithm supports improvements in speedup and resistance to side-channel attacks (SCA), as mentioned in the Introduction. Therefore, the speed up has to be compared with other existing proposals, not only with the original algorithm. Furthermore, the SCA resistance aspect should be further analyzed.

  • The authors should explain the targeted applications of these proposals, then prove that the experimental system and results are appropriate.

Author Response

(The authors gave the same response as above.)

Round 2

Reviewer 2 Report

The authors have addressed all my concerns.

Reviewer 3 Report

The authors have fixed all the previous major issues.

There is only one thing left. Figures 5, 6, 7, and 8 are still too small to read.

Please enlarge them before publishing.